# Association of Adiposity with Periodontitis and Metabolic Syndrome: From the Third National Health and Nutrition Examination Survey of United States

**DOI:** 10.3390/ijerph20032533

**Published:** 2023-01-31

**Authors:** YunSook Jung, Ji-Hye Kim, Ah-Ra Shin, Keun-Bae Song, Atsuo Amano, Youn-Hee Choi

**Affiliations:** 1Department of Dental Hygiene, College of Science & Technology, Kyungpook National University, Sangju 37224, Republic of Korea; 2Department of Preventive Dentistry, School of Dentistry, Kyungpook National University, Daegu 41940, Republic of Korea; 3Craniofacial Nerve-Bone Network Research Center, Kyungpook National University School of Dentistry, Daegu 41940, Republic of Korea; 4Department of Preventive Dentistry, Osaka University Graduate School of Dentistry, Osaka 565-0871, Japan; 5Institute for Translational Research in Dentistry, Kyungpook National University, Daegu 41940, Republic of Korea

**Keywords:** adiposity, metabolic syndrome, periodontitis

## Abstract

This study explored the epidemiological role of central adiposity and body mass index (BMI) in terms of clinical attachment loss (CAL)/pocket depth (PD) and metabolic syndrome components. This study included data from the National Health and Nutrition Examination Survey III of America on 12,254 adults aged 20 years of age or older with a blood sample, anthropometric measurements, and a periodontal examination. Clinical periodontitis measurements, including CAL and PD, were classified into quintiles or quartiles and compared. CAL was positively associated with central adiposity, hypertension, and hyperglycemia; the relationship between CAL and diabetes was stronger when central adiposity was absent (odds ratio [OR] and 95% confidence interval: 6.33, 2.14–18.72 vs. 3.14, 1.78–5.56). The relationship between CAL and impaired fasting glucose (IFG) differed slightly with BMI. The IFG ORs for normal, overweight, and obese patients were 1.63 (1.08–2.45), 1.76 (1.05–2.97), and 1.43 (0.88–2.30), respectively. CAL was positively correlated with all metabolic syndrome components except hypertriglyceridemia. Associations between CAL, diabetes, and IFG significantly varied with BMI. Periodontitis in individuals without central obesity or with normal bodyweight may independently indicate diabetes and IFG. Therefore, preventive measures against periodontitis without obesity are necessary to improve general and oral health.

## 1. Introduction

Epidemiological evidence suggests that metabolic syndrome, a major global public health challenge [1], is associated with periodontitis [2,3,4,5,6]. Those exhibiting more components of metabolic syndrome are at a significantly higher risk of developing periodontitis. Most studies to date have focused on the association between individual components of metabolic syndrome, such as dyslipidemia [7,8,9], high blood pressure [10,11], hyperglycemia [12,13], and central adiposity or obesity [14,15,16,17,18,19] and periodontitis.

A recent cohort study found that the presence of periodontal pockets was associated with the development of one or more metabolic syndrome components, suggesting that efforts to prevent periodontal disease could also potentially minimize the risk of developing metabolic syndrome [4,20]. Some studies have indicated a positive association between insulin resistance and periodontal disease [5,21,22]; in contrast, one study demonstrated a significant association between metabolic syndrome and periodontal disease in women but not in men [23].

Over the last two decades, several attempts have been made to understand the biological mechanisms underlying this association between periodontal and metabolic diseases [17,24,25], including studies that explored the role of adipose tissue in the homeostatic process as it relates to obesity, which can lead to hemodynamic, neurohormonal, and metabolic alterations [26,27].

Adipocytokines may play an essential role in the relationship between hyperglycemia, obesity, and periodontitis [28,29,30]. Oxidative stress associated with periodontitis and metabolic syndrome is suggested to modulate adipocytokines and impact glucose homeostasis [31]. After adjusting for glucose tolerance, periodontal disease was found to be positively associated with obesity [14], whereas body mass index (BMI) was positively associated with periodontitis in patients with type 2 diabetes mellitus [32]. Periodontal disease is also associated with metabolic syndrome components [33,34,35,36]. However, the relationship with adiposity needs to be studied using epidemiological methods.

The current study examined the association between periodontal disease measurements, including clinical attachment loss (CAL) and pocket depth (PD), with metabolic syndrome components and explored the epidemiological role of central and total body adiposity in the relationship between periodontitis, hyperglycemia, and other metabolic syndrome components.

## 2. Materials and Methods

### 2.1. Study Population

This study included data on adults 20 years of age or older and included in the National Health and Nutrition Examination Survey (NHANES) III in the United States, resulting in 12,254 individuals with available blood test results, anthropometric measurements, and a recent history of periodontal examination being included. Further details of the inclusion and exclusion criteria for the original NHANES datasets have been reported previously [34]. Publicly available NHANES datasets have been previously approved by the National Center for Health Statistics Institutional Review Board, Center for Disease Control and Prevention; thus, no further internal ethical review was necessary. This observational study was conducted in accordance with the Strengthening the Reporting of Observational Studies in Epidemiology (STROBE) statement.

### 2.2. Periodontal Measurement

Clinical assessment of adult chronic periodontal inflammation included measurement of CAL and PD, and the study participants were classified into four or five groups based on these measurements. The mean CAL and PD per person were calculated and the measurements were categorized into the first quintile/quartile through to the fifth quintile/fourth quartile, with the first quintile/quartile being used as the reference category. Currently, no unanimously accepted clinical definition of adult chronic periodontitis exists. We intended to reduce the misclassification bias of measurements of periodontal exposure [37].

### 2.3. Metabolic Syndrome Components

In this study, metabolic syndrome was as defined in the National Cholesterol Education Program’s Adult Treatment Panel III report [38]: (1) increased waist circumference ≥102 cm (40 inches) in men and ≥88 cm (35 inches) in women; (2) elevated triglyceride levels (≥150 mg/dL [1.7 mmol/L]); (3) decreased levels of high-density lipoprotein (HDL) cholesterol (≤40 mg/dL [1.03 mmol/L] for men and ≤50 mg/dL [1.3 mmol/L] for women); (4) elevated blood pressure (systolic, ≥130 mmHg or diastolic, ≥85 mmHg); and (5) elevated fasting blood glucose (≥110 mg/dL). The American Diabetes Association [39] reduced the fasting plasma glucose-level cutoff from 110 mg/dL (6.1 mmol/L) to 100 mg/dL (5.6 mmol/L); however, the previous cutoff was used in this study.

### 2.4. Confounding Covariates

Critical confounding factors, including age, sex, race/ethnicity, income-to-poverty ratio, years of education, general health indicators such as BMI, smoking history, drinking habits, and physical activity, and oral health behaviors such as frequency of dental visits per year, were considered.

BMI was determined as an index of total body obesity by dividing the patient’s weight in kilograms by height in meters squared (kg/m^2^). Leisure-time physical activity was evaluated using questions focusing on the frequency and type of activity and reported as units of metabolic equivalents. Finally, the number of missing teeth was recorded as another potential confounder for periodontitis. All covariates were categorized using published previously methods [34].

### 2.5. Statistical Analysis

NHANES III used a complex, stratified, and multistage sampling design and was conducted by the National Center for Health Statistics (Hyattsville, MD, USA). SAS (version 9.1, SAS Institute Inc., Carey, NC, USA) survey procedures with weighting were used to consider national sampling, resulting in less bias in the strengths of association and standard errors observed.

To describe the characteristics of the study population, the mean values of CAL, PD, and the components of metabolic syndrome were calculated by sociodemographic factors and other potential risk factors. Logistic regression models that took the CAL and PD categories, dichotomized clinical measures of each metabolic syndrome component, and all confounding factors into consideration were used to examine the association between periodontal disease and metabolic syndrome. A linear test for a trend across the CAL and PD categories was also conducted.

To evaluate the effect modifications between periodontal disease, central adiposity, and components of metabolic syndrome, the logistic regression models including periodontal disease measures and metabolic syndrome components were initially stratified by central adiposity. Next, multiplicative terms between the categories of periodontal disease and central adiposity were introduced in the multivariate models to evaluate joint associations between categories. This approach was repeated to evaluate joint associations between periodontal disease and central and general adiposity in relation to diabetes outcomes and impaired fasting glucose (IFG).

## 3. Results

The weighted average values and standard errors of CAL, PD, and five metabolic syndrome components in terms of sociodemographic and other potential risk factors were reported. Individuals who were younger, more educated, had more teeth, underwent regular dental checkups, had never smoked, and were less obese tended to have low CAL and PD and more favorable levels of metabolic syndrome components (Table 1).

Significant linear associations between CAL and PD and individual metabolic syndrome components, excluding hypertriglyceridemia, were observed (Table 2). CAL was positively associated with central adiposity, hypertension, and hyperglycemia (*p* < 0.05). The odds ratio (OR) of the fifth quintile of CAL was significantly higher for HDL (OR: 1.20). This pattern was considerably weaker when PD was used to indicate periodontal damage. The ORs of the fourth and fifth quintiles of CAL were statistically significant for central adiposity, and the OR of the fourth quintile was significant for hypertension. Moreover, the linearity trend of the ORs was also statistically significant.

Examination of the effect modifications showed that the relationship between CAL, hypertension, and hyperglycemia was slightly stronger in those without central adiposity (Table 3). However, these relationships remained unclear when using PD as a measure of periodontitis.

Table 4 presents the effect modification of obesity on the visceral and total body for diabetes and IFG according to CAL and PD quartiles. The association between CAL and diabetes was considerably stronger in individuals without central adiposity compared with those with central adiposity (*p* = 0.10). Similarly, the ORs for the association between CAL and diabetes were substantially higher in individuals with normal body weight compared with those who were overweight or obese (adjusted OR for diabetes across extreme quartiles of CAL: 4.96 vs. 2.81 and 4.42, respectively). Although these patterns were also observed for IFG, they were considerably weaker (for extreme quartiles of CAL: 1.63 vs. 1.76 and 1.43, respectively). No obvious relationships with PD were observed.

## 4. Discussion

This study included a representative sample of adults in the US. The results of this study revealed that the associations between the clinical measurements of periodontitis, CAL, hypertension, and hyperglycemia were stronger in the absence of central adiposity after adjusting for potential confounding factors.

Hypertension is more prevalent among individuals with obesity or diabetes [40]. It has been suggested that the underlying biological link is a consequence of adipose tissue function, which affects the immune response, blood pressure control, and glucose homeostasis [27]. There is considerable empirical evidence supporting the association between periodontitis and hyperglycemia [13], hypertension [10,11,29,34], and obesity [14,15,16,17,18], suggesting that periodontitis plays a significant role in the complexity of linked biomechanisms [31]. The results of the current study provide substantial evidence in support of this.

This study has a few essential implications. First, as CAL measures periodontal disease over a prolonged period, it may a better predictor of the associations between periodontitis and the components of metabolic syndrome than PD, despite the lack of a standardized definition when measuring clinical periodontal inflammation. Second, classifying CAL and PD into quintiles (or quartiles) may increase the detection of associations. The prevalence of periodontitis can vary depending on its definition. For example, when periodontitis is defined as the presence of at least one site with a CAL of ≥2 mm, approximately 80% of all adults can be considered affected, with approximately 90% of these aged 55–64 years [41]. When it is defined as a PD of ≥4 mm, 30% of the adults were found to exhibit at least three to four teeth that met the criterion. Third, the cross-sectional associations between CAL, diabetes, and IFG were likely to be slightly stronger in non-obese individuals. The biological rationale for this finding could be that this group of lean individuals with diabetes included those who had had the disease for a long time and lost weight after diagnosis. However, further studies are warranted to confirm this. Despite these considerations, the findings of this study can still be useful. When examining patients with periodontitis despite having normal body weight and no central obesity, dentists should consider the possibility of diabetes and IFG. To the best of our knowledge, these findings have not been reported previously.

However, this study also had several limitations. First, this was a cross-sectional study that used the NHANES III dataset; therefore, the causality of periodontitis and metabolic syndrome could not be elucidated. However, the findings provide epidemiological evidence for a representative adult population in the US. Second, comorbidity could not be considered as a component of metabolic. To accurately understand the relationship between CAL/PD and the components of metabolic syndrome, inclusion of other metabolic syndrome components in the same model may be necessary. However, CAL may still be considered a significant predictor of visceral obesity, hypertension, and hyperglycemia despite this limitation. Third, the high ORs and wide confidence intervals (e.g., in individuals without central adiposity or with normal BMI, the adjusted ORs of CAL for diabetes were 6.33 [2.14–18.72] and 4.96 [1.23–20.09], respectively) shown in Table 4 suggest lack of precision, potentially due to small cell numbers in the subgroup analyses following stratification. Fourth, several recently used indicators were not available because of the use of old provided data; an interesting finding of association between blood type and chronic periodontitis has been reported recently [42]. The present study did not consider the blood group as a confounder. This is one of the limitations of this study. The advantage of using ABO blood groups for further studies needs to be included. Despite this limitation, the findings indicated a distinct association. Finally, the effects of multiple testing must be taken into consideration when calculating the ORs of periodontitis and diabetes in subpopulations stratified using the effect modifiers in this study.

## 5. Conclusions

In conclusion, periodontal disease was positively associated with central adiposity, dyslipidemia, hypertension, and hyperglycemia. Higher loss of periodontal attachment was more strongly associated with diabetes and IFG in the absence of adiposity. However, further studies are necessary to elucidate the underlying biological mechanisms. The findings of the current study may help dentists and physicians identify patients with undiagnosed diabetes and periodontitis.

## Figures and Tables

**Table 1 ijerph-20-02533-t001:** The levels of periodontal measures and metabolic syndrome components by characteristics of the study population.

Variables	N	Periodontitis		Metabolic Syndrome Components					
		Clinical Attachment Loss (mm)	Pocket Depth (mm)	Waist Circumference (cm)	Triglyceride (mg/dL)	High-Density Lipoprotein (mg/dL)	Systolic Blood Pressure (mmHg)	Diastolic Blood Pressure (mmHg)	Fasting Glucose (mg/dL)
									Mean (SE)
Total	12,254	1.09 (0.03)	1.47 (0.02)	91.30 (0.25)	139.07 (2.25)	50.51 (0.35)	120.85 (0.37)	74.30 (0.20)	97.43 (0.41)
Age (years)									
18–44	7135	0.76 (0.03)	1.45 (0.02)	88.54 (0.32)	125.64 (2.62)	50.12 (0.41)	115.06 (0.28)	72.96 (0.24)	93.13 (0.51)
45–64	3044	1.56 (0.05)	1.51 (0.03)	96.11 (0.37)	164.60 (4.31)	50.96 (0.55)	126.96 (0.41)	77.73 (0.27)	103.11 (0.74)
65 and over	2075	1.98 (0.07)	1.47 (0.03)	96.65 (0.46)	159.35 (4.01)	51.77 (0.69)	140.65 (0.58)	74.26 (0.38)	109.83 (1.17)
Sex									
Male	5920	1.20 (0.04)	1.55 (0.02)	94.90 (0.26)	153.74 (2.93)	45.87 (0.39)	123.45 (0.40)	76.63 (0.24)	99.32 (0.56)
Female	6334	0.99 (0.04)	1.39 (0.02)	87.70 (0.37)	124.32 (2.69)	55.16 (0.42)	118.22 (0.44)	71.97 (0.23)	95.54 (0.57)
Race									
Non-Hispanic White	4617	1.07 (0.04)	1.43 (0.03)	91.40 (0.31)	141.28 (2.78)	50.31 (0.43)	120.91 (0.47)	74.23 (0.23)	97.03 (0.54)
Non-Hispanic Black	3446	1.23 (0.04)	1.67 (0.03)	91.91 (0.35)	109.23 (1.58)	54.85 (0.41)	123.04 (0.38)	75.42 (0.29)	98.85 (0.81)
Mexican-American	3677	0.97 (0.05)	1.56 (0.02)	92.16 (0.41)	156.24 (3.07)	48.12 (0.41)	119.46 (0.41)	73.41 (0.41)	99.83 (0.61)
Others	514	1.18 (0.06)	1.51 (0.03)	88.98 (0.86)	145.97 (5.63)	48.23 (1.01)	118.38 (0.99)	74.03 (0.53)	97.57 (1.37)
Education level (years)									
≤6	1672	1.66 (0.09)	1.68 (0.03)	93.93 (0.75)	155.09 (5.37)	48.26 (0.81)	125.10 (1.33)	74.22 (0.44)	103.89 (1.96)
7–12	6087	1.20 (0.04)	1.52 (0.02)	92.34 (0.32)	142.65 (3.12)	49.77 (0.45)	121.70 (0.41)	74.16 (0.28)	97.58 (0.60)
≥13	3820	0.89 (0.04)	1.38 (0.02)	89.82 (0.34)	131.42 (2.98)	51.52 (0.47)	119.09 (0.47)	74.40 (0.28)	95.84 (0.66)
Income poverty ratio ^a^									
Lower (≤1.5)	4088	1.20 (0.05)	1.59 (0.03)	91.49 (0.50)	139.48 (3.31)	49.79 (0.54)	119.91 (0.47)	73.32 (0.26)	98.21 (0.65)
Middle (≤3.0)	3536	1.13 (0.04)	1.51 (0.02)	91.73 (0.50)	143.16 (3.60)	49.30 (0.49)	120.34 (0.46)	73.96 (0.25)	97.68 (0.69)
Higher (>3.0)	3564	1.01 (0.03)	1.39 (0.02)	90.92 (0.35)	136.15 (3.32)	51.57 (0.51)	121.36 (0.50)	75.00 (0.29)	96.54 (0.63)
No. of missing teeth ^b^									
0	5667	0.89 (0.03)	1.41 (0.02)	89.24 (0.34)	130.80 (2.47)	51.52 (0.45)	118.95 (0.43)	73.53 (0.23)	94.96 (0.39)
1–5	5037	1.19 (0.04)	1.51 (0.02)	92.89 (0.41)	144.40 (2.91)	49.64 (0.38)	122.03 (0.43)	74.85 (0.29)	98.78 (0.75)
6–10	1155	1.80 (0.08)	1.61 (0.05)	97.14 (0.75)	170.78 (9.03)	47.99 (0.69)	126.42 (0.81)	76.57 (0.36)	106.86 (1.87)
11+	395	2.65 (0.19)	1.87 (0.09)	101.28 (1.63)	162.97 (10.86)	46.71 (1.45)	134.25 (1.43)	78.32 (0.72)	110.95 (4.21)
Smoking									
Never smoker	6215	0.87 (0.03)	1.39 (0.02)	90.14 (0.33)	131.19 (3.28)	51.90 (0.47)	120.24 (0.40)	74.06 (0.22)	96.87 (0.59)
Ex-smoker	2817	1.30 (0.04)	1.46 (0.02)	95.25 (0.49)	154.44 (3.51)	49.96 (0.41)	124.84 (0.64)	76.03 (0.38)	101.71 (0.91)
Current smoker	3222	1.29 (0.05)	1.60 (0.03)	89.82 (0.40)	139.02 (2.47)	48.66 (0.44)	118.43 (0.45)	73.21 (0.33)	94.69 (0.67)
Alcohol									
Never	1984	1.12 (0.05)	1.44 (0.02)	90.23 (0.54)	140.69 (4.05)	50.39 (0.55)	121.09 (0.66)	72.93 (0.32)	99.73 (1.06)
Former	4025	1.24 (0.04)	1.49 (0.03)	93.84 (0.40)	153.08 (4.38)	47.54 (0.43)	122.36 (0.42)	74.57 (0.28)	100.53 (0.70)
Current	6002	1.01 (0.04)	1.46 (0.02)	90.31 (0.28)	131.68 (2.23)	52.03 (0.43)	120.07 (0.41)	74.46 (0.25)	95.43 (0.53)
Physical activity ^c^									
Active	6115	1.03 (0.04)	1.46 (0.02)	91.67 (0.30)	141.31 (3.36)	50.23 (0.44)	120.36 (0.40)	74.44 (0.22)	97.10 (0.61)
Moderate	1254	1.09 (0.06)	1.42 (0.03)	88.98 (0.54)	135.45 (4.61)	52.58 (0.65)	122.26 (0.83)	74.55 (0.48)	96.43 (0.60)
Less active	924	0.94 (0.04)	1.41 (0.03)	89.08 (0.76)	125.48 (4.55)	50.25 (0.88)	119.88 (0.80)	73.50 (0.50)	95.38 (0.70)
Unknown	3961	1.30 (0.04)	1.52 (0.02)	92.38 (0.55)	140.50 (2.87)	50.23 (0.49)	121.65 (0.44)	74.14 (0.37)	99.45 (0.70)
Central adiposity									
Yes	4859	1.31 (0.04)	1.54 (0.02)	105.53 (0.30)	176.74 (3.38)	46.90 (0.34)	126.88 (0.41)	77.28 (0.25)	104.95 (0.77)
No	6979	0.97 (0.04)	1.43 (0.02)	83.70 (0.21)	118.10 (2.25)	52.52 (0.46)	117.45 (0.38)	72.70 (0.23)	93.27 (0.46)
BMI (kg/m^2^)									
<18.5	227	0.98 (0.08)	1.41 (0.05)	69.46 (0.39)	83.03 (3.99)	57.59 (1.25)	109.98 (1.08)	68.47 (0.61)	88.39 (0.76)
<25.0	4566	0.97 (0.04)	1.40 (0.03)	80.57 (0.18)	104.67 (1.90)	55.25 (0.46)	116.80 (0.51)	71.52 (0.27)	92.31 (0.42)
<30.0	4281	1.17 (0.04)	1.49 (0.02)	94.61 (0.15)	157.41 (3.00)	47.72 (0.36)	123.02 (0.49)	75.75 (0.24)	99.05 (0.83)
≥30.0	3160	1.24 (0.04)	1.56 (0.03)	109.75 (0.35)	184.14 (4.25)	44.69 (0.46)	126.57 (0.38)	78.21 (0.30)	105.95 (1.08)

^a^ Income Poverty ratio: (Midpoint family income)/(poverty threshold values based on calendar years and inflation), ^b^ Missing due to caries/periodontal disease, ^c^ The number of missing values was 4097 categorized by unknown, At least once a year visit to dental clinic, cf. Numbers and medians: Unweighted value, Means: Weighted value.

**Table 2 ijerph-20-02533-t002:** Association between periodontitis and metabolic syndrome components.

Total (N = 12,254)	Quintiles for Clinical Attachment Loss					Quintiles for Pocket Depth				
	Q1 (n = 2412)	Q2 (n = 2493)	Q3 (n = 2465)	Q4 (n = 2431)	Q5 (n = 2453)	Q1 (n = 2451)	Q2 (n = 2413)	Q3 (n = 2512)	Q4 (n = 2429)	Q5 (n = 2449)
Mean waist circumference (cm)	86.84	89.56	91.42	94.24	96.55	88.08	89.79	91.88	93.30	96.36
Central adiposity * (%)	23.86	30.19	35.04	41.59	48.94	29.89	30.59	34.76	39.47	45.48
OR (95% CI)	1.00	1.10 (0.74–1.63)	1.52 (1.05–2.21)	1.46 (0.96–2.22)	1.82 (1.07–3.12)	1.00	1.08 (0.83–1.40)	1.18 (0.81–1.70)	1.91 (1.35–2.70)	1.91 (1.20–3.04)
*p*-value for trend	0.015					*0.001*				
Mean triglycerides (mg/dL)	125.55	132.62	132.95	152.20	160.19	130.35	137.68	139.23	140.51	155.13
Hypertriglyceridemia ^†^ (%)	20.89	27.42	28.94	35.34	39.69	25.37	28.36	30.16	31.57	36.68
OR (95% CI)	1.00	1.21 (0.98–1.50)	1.06 (0.81–1.38)	1.15 (0.88–1.51)	1.09 (0.79–1.50)	1.00	1.10 (0.92–1.31)	1.00 (0.83–1.19)	1.08 (0.86–1.36)	1.01 (0.79–1.29)
*p*-value for trend	*0.729*					*0.950*				
Mean high-density lipoprotein (mg/dL)	52.01	49.94	50.82	50.23	49.03	52.70	50.75	49.91	50.03	47.70
Dyslipidemia ^‡^ (%)	31.43	36.67	35.89	36.98	42.71	32.71	36.05	35.90	36.10	44.13
OR (95% CI)	1.00	1.13 (0.91–1.39)	1.01 (0.78–1.31)	1.01 (0.80–1.27)	1.20 (1.00–1.45)	1.00	1.18 (0.97–1.42)	1.04 (0.82–1.31)	1.02 (0.84–1.23)	1.21 (0.98–1.49)
*p*-value for trend	*0.473*					*0.419*				
Mean systolic blood pressure (mmHg)	115.04	117.25	119.99	124.71	130.79	119.14	119.79	120.94	121.93	124.25
Mean diastolic blood pressure (mmHg)	71.83	73.87	74.72	75.73	76.11	73.15	74.07	74.67	74.77	75.59
Hypertension ^§^ (%)	13.49	21.72	28.31	37.70	49.16	25.02	25.08	28.68	32.56	35.60
OR (95% CI)	1.00	1.53 (1.17–1.99)	1.70 (1.31–2.20)	1.80 (1.42–2.29)	1.80 (1.33–2.44)	1.00	1.05 (0.84–1.31)	1.13 (0.94–1.37)	1.44 (1.19–1.75)	1.24 (0.96–1.59)
*p*-value for trend	*<0.001*					*0.001*				
Mean fasting glucose (mg/dL)	91.21	93.87	96.59	101.71	107.54	94.97	96.89	96.85	997.54	103.57
Hyperglycemia ^||^ (%)	12.91	16.01	22.33	29.92	38.62	20.23	20.13	23.42	22.30	31.64
OR (95% CI)	1.00	1.07 (0.77–1.47)	1.43 (1.09–1.88)	1.67 (1.23–2.26)	1.84 (1.41–2.39)	1.00	0.94 (0.73–1.22)	1.04 (0.81–1.33)	0.93 (0.74–1.16)	1.32 (0.98–1.79)
*p*-value for trend	*<0.001*					*0.133*				

Adjusted for age, sex, race, education, income, missing teeth, smoking, alcohol intake, regularity of dental visits, BMI, and physical activity. * Central adiposity: waist circumference > 102 cm (men) or >88 cm (women). ^†^ Hypertriglyceridemia: triglyceride ≥ 150 mg/dL. ^‡^ Dyslipidemia: HDL cholesterol < 40 mg/dL (men) or <50 mg/dL (women). ^§^ Hypertension: blood pressure ≥ 130/85 mmHg. ^||^ Hyperglycemia: fasting glucose ≥ 100 mg/dL.

**Table 3 ijerph-20-02533-t003:** Joint association between periodontitis and central adiposity in relation to metabolic syndrome components.

Total (N = 12,254)	Clinical Attachment Loss (OR, 95% CI)					Pocket Depth (OR, 95% CI)				
	Q1 (n = 2412)	Q2 (n = 2493)	Q3 (n = 2465)	Q4 (n = 2431)	Q5 (n = 2453)	Q1 (n = 2451)	Q2 (n = 2413)	Q3 (n = 2512)	Q4 (n = 2429)	Q5 (n = 2449)
Hypertriglyceridemia										
Stratification										
Central adiposity absent	1.00	1.26 (0.90–1.77)	0.86 (0.60–1.22)	1.16 (0.78–1.73)	1.52 (0.83–1.89)	1.00	1.03 (0.79–1.34)	0.94 (0.73–1.21)	1.20 (0.89–1.94)	1.23 (0.84–1.80)
Central adiposity present	1.00	1.17 (0.88–1.55)	1.21 (0.86–1.69)	1.02 (0.73–1.43)	0.91 (0.65–1.28)	1.00	1.18 (0.91–1.53)	0.97 (0.74–1.29)	0.87 (0.61–1.23)	0.76 (0.56–1.04)
*Joint classification*										
Central adiposity absent	1.00	1.26 (0.90–1.77)	0.86 (0.60–1.22)	1.16 (0.78–1.73)	1.52 (0.83–1.89)	1.00	1.03 (0.79–1.34)	0.94 (0.73–1.21)	1.20 (0.89–1.94)	1.23 (0.84–1.80)
Central adiposity present	1.60 (1.03–2.48)	1.87 (1.23–2.84)	1.92 (1.30–2.86)	1.63 (1.04–2.55)	1.45 (0.88–2.38)	1.69 (1.16–2.45)	1.99 (1.36–2.92)	1.64 (1.21–2.24)	1.46 (0.93–2.30)	1.28 (0.83–1.98)
Dyslipidemia										
*Stratification*										
Central adiposity absent	1.00	0.99 (0.74–1.33)	0.88 (0.61–1.25)	0.89 (0.67–1.19)	1.22 0.94–1.58)	1.00	1.09 (0.83–1.42)	0.98 (0.71–1.35)	0.89 (0.69–1.16)	1.22 (0.93–1.61)
Central adiposity present	1.00	1.39 (0.97–2.00)	1.22 (0.92–1.62)	1.25 (0.81–1.93)	1.28 (0.93–1.76)	1.00	1.32 (0.95–1.82)	1.10 (0.77–1.57)	1.11 (0.76–1.62)	1.16 (0.86–1.58)
*Joint classification*										
Central adiposity absent	1.00	0.99 (0.74–1.33)	0.88 (0.61–1.25)	0.89 (0.67–1.19)	1.22 (0.94–1.58)	1.00	1.09 (0.83–1.42)	0.98 (0.71–1.35)	0.89 (0.69–1.16)	1.22 (0.93–1.61)
Central adiposity present	1.30 (0.90–1.86)	1.81 (1.14–2.88)	1.58 (1.13–2.21)	1.62 (1.16–2.28)	1.66 (1.19–2.31)	1.47 (0.96–2.26)	1.93 (1.36–2.76)	1.61 (1.05–2.48)	1.63 (1.16–2.28)	1.71 (1.14–2.55)
Hypertension										
*Stratification*										
Central adiposity absent	1.00	1.68 (1.12–2.51)	1.76 (1.17–2.64)	1.94 (1.38–2.74)	2.01 (1.41–2.87)	1.00	1.03 (0.74–1.44)	0.51 (0.72–1.40)	1.34 (0.99–1.82)	1.08 (0.74–1.58)
Central adiposity present	1.00	1.42 (1.01–2.00)	1.54 (1.09–2.17)	1.56 (1.00–2.43)	1.57 (1.03–2.38)	1.00	1.07 (0.74–1.54)	1.26 (0.95–1.68)	1.44 (0.97–2.13)	1.41 (1.01–1.98)
*Joint classification*										
Central adiposity absent	1.00	1.68 (1.12–2.51)	1.76 (1.17–2.64)	1.94 (1.38–2.74)	2.01 (1.41–2.87)	1.00	1.03 (0.74–1.44)	0.51 (0.72–1.40)	1.34 (0.99–1.82)	1.08 (0.74–1.58)
Central adiposity present	1.91 (1.18–3.08)	2.72 (1.65–4.50)	2.94 (1.88–4.59)	2.98 (1.87–4.75)	2.99 (1.92–4.66)	1.44 (0.97–2.12)	1.54 (1.08–2.18)	1.81 (1.26–2.59)	2.06 (1.40–3.03)	2.03 (1.41–2.90)
Hyperglycemia										
*Stratification*										
Central adiposity absent	1.00	1.10 (0.77–1.56)	1.44 (1.03–2.00)	1.82 (1.28–2.59)	1.88 (1.31–2.70)	1.00	0.75 (0.53–1.05)	0.96 (0.69–1.34)	0.83 (0.60–1.15)	1.41 (0.93–2.13)
Central adiposity present	1.00	0.90 (0.59–1.37)	1.30 (0.82–2.05)	1.47 (0.93–2.33)	1.64 (1.12–2.42)	1.00	1.18 (0.86–1.62)	1.08 (0.82–1.43)	1.05 (0.75–1.48)	1.21 (0.87–1.68)
*Joint classification*										
Central adiposity absent	1.00	1.10 (0.77–1.56)	1.44 (1.03–2.00)	1.82 (1.28–2.59)	1.88 (1.31–2.70)	1.00	0.75 (0.53–1.05)	0.96 (0.69–1.34)	0.83 (0.60–1.15)	1.41 (0.93–2.13)
Central adiposity present	1.79 (1.11–2.88)	1.62 (1.01–2.58)	2.33 (1.59–3.40)	2.64 (1.68–4.14)	2.94 (2.02–4.29)	1.38 (1.03–1.84)	1.63 (1.16–2.29)	1.49 (1.03–2.16)	1.45 (1.03–2.03)	1.67 (1.19–2.36)

**Table 4 ijerph-20-02533-t004:** Joint association between periodontitis and central adiposity in relation to diabetes and impaired fasting glucose by CAL and PD quartiles.

Total (N = 12,254)	Clinical Attachment Loss (OR, 95% CI)				Pocket Depth (OR, 95% CI)			
	Q1 (n = 3113)	Q2 (n = 3015)	Q3 (n = 3066)	Q4 (n = 3060)	Q1 (n = 3058)	Q2 (n = 2935)	Q3 (n = 3163)	Q4 (n = 3098)
**Diabetes**								
*Stratification*								
Central adiposity absent	1.00	2.17 (0.64–7.34)	2.99 (1.15–7.76)	6.33 (2.14–18.72)	1.00	1.26 (0.63–2.53)	1.48 (0.66–3.34)	2.41 (1.18–4.89)
Central adiposity present	1.00	1.98 (1.11–3.55)	2.03 (1.24–3.32)	3.14 (1.78–5.56)	1.00	1.10 (0.68–1.78)	1.40 (0.86–2.30)	1.30 (0.84–2.01)
*Joint classification*								
Central adiposity absent	1.00	2.17 (0.64–7.34)	2.99 (1.15–7.76)	6.33 (2.14–18.72)	1.00	1.26 (0.63–2.53)	1.48 (0.66–3.34)	2.41 (1.18–4.89)
Central adiposity present	4.15 (1.54–11.21)	8.22 (2.95–22.91)	8.40 (3.55–19.87)	13.04 (5.33–31.89)	3.43 (1.69–6.97)	3.77 (2.01–7.10)	4.81 (2.68–8.65)	4.45 (2.36–8.39)
Diabetes								
*Stratification*								
Normal weight	1.00	2.19 (0.46–10.46)	2.33 (0.77–7.08)	4.96 (1.23–20.09)	1.00	1.22 (0.57–2.62)	1.13 (0.43–2.98)	2.08 (0.84–5.15)
Overweight	1.00	2.05 (0.83–5.03)	1.92 (0.87–4.23)	2.81 (1.10–7.19)	1.00	0.91 (0.54–1.53)	1.99 (1.02–3.88)	1.56 (0.91–2.68)
Obese	1.00	2.14 (1.09–4.21)	2.58 (1.57–4.24)	4.42 (2.36–8.28)	1.00	1.28 (0.69–2.36)	1.22 (0.68–2.19)	1.39 (0.85–2.27)
*Joint classification*								
Normal weight	1.00	2.19 (0.46–10.46)	2.33 (0.77–7.08)	4.96 (1.23–20.09)	1.00	1.22 (0.57–2.62)	1.13 (0.43–2.98)	2.08 (0.84–5.15)
Overweight	1.78 (0.38–8.32)	3.63 (0.69–19.13)	3.42 (0.96–12.23)	5.00 (1.17–21.28)	1.25 (0.55–2.84)	1.13 (0.44–2.90)	2.48 (0.98–6.27)	1.94 (0.77–4.93)
Obese	2.88 (0.58–14.16)	6.16 (1.34–28.40)	7.42 (1.57–34.99)	12.71 (2.94–55.00)	2.86 (1.10–7.46)	3.66 (1.28–10.45)	3.48 (1.33–9.14)	3.98 (1.60–9.89)
IFG								
*Stratification*								
Central adiposity absent	1.00	1.42 (1.01–1.99)	1.65 (1.25–2.19)	1.72 (1.16–2.54)	1.00	0.83 (0.62–1.13)	1.03 (0.77–1.37)	1.26 (0.84–1.90)
Central adiposity present	1.00	0.98 (0.68–1.40)	1.22 (0.76–1.95)	1.27 (0.87–1.85)	1.00	0.98 (0.75–1.29)	1.22 (0.90–1.66)	1.20 (0.90–1.59)
Joint classification								
Central adiposity absent	1.00	1.42 (1.01–1.99)	1.65 (1.25–2.19)	1.72 (1.16–2.54)	1.00	0.83 (0.62–1.13)	1.03 (0.77–1.37)	1.26 (0.84–1.90)
Central adiposity present	1.86 (1.18–2.94)	1.81 (1.25–2.64)	2.26 (1.49–3.43)	2.36 (1.64–3.40)	1.32 (0.96–1.81)	1.29 (0.95–1.76)	1.61 (1.10–2.37)	1.58 (1.17–2.15)
IFG								
*Stratification*								
Normal weight	1.00	0.86 (0.58–1.28)	1.31 (0.89–1.95)	1.63 (1.08–2.45)	1.00	0.66 (0.43–1.02)	0.80 (0.53–1.20)	1.04 (0.70–1.55)
Overweight	1.00	2.18 (1.35–3.53)	1.99 (1.32–2.99)	1.76 (1.05–2.97)	1.00	0.98 (0.66–1.45)	1.25 (0.81–1.92)	1.25 (0.75–2.06)
Obese	1.00	0.91 (0.50–1.65)	1.29 (0.69–2.42)	1.43 (0.88–2.30)	1.00	1.28 (0.80–2.04)	1.52 (1.00–2.31)	1.64 (1.09–2.49)
*Joint classification*								
Normal weight	1.00	0.86 (0.58–1.28)	1.31 (0.89–1.95)	1.63 (1.08–2.45)	1.00	0.66 (0.43–1.02)	0.80 (0.53–1.20)	1.04 (0.70–1.55)
Overweight	0.92 (0.60–1.41)	2.00 (1.35–2.97)	1.83 (1.30–2.58)	1.62 (1.05–2.51)	1.07 (0.75–1.54)	1.05 (0.73–1.52)	1.34 (0.90–1.99)	1.34 (0.81–2.20)
Obese	1.47 (0.85–2.55)	1.33 (0.79–2.24)	1.90 (1.03–3.53)	2.10 (1.18–3.73)	0.93 (0.51–1.67)	1.18 (0.62–2.26)	1.41 (0.88–2.24)	1.52 (0.88–2.64)

## Data Availability

The data used for this analysis are available at: https://wwwn.cdc.gov/nchs/nhanes/search/datapage.aspx?Component=examination, (accessed on 28 December 2022).

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
