# Peer review of "Association of Adiposity with Periodontitis and Metabolic Syndrome: From the Third National Health and Nutrition Examination Survey of United States"

_ijerph, 2023, doi:10.3390/ijerph20032533_

Round 1

Reviewer 1 Report

This study studied the association of periodontal measurements of clinical attachment loss (CAL) and pocket depth (PD) with metabolic syndrome components in US subjects. Some important comments are as follows.

It is better to specify the age range rather than mentioning aged ≥20 years.

Please mention the details of the study years?

The authors mention that they used the date from the National Health and Nutrition Examination Survey (NHANES) III in the United States. Did they obtain the approval?

In the statistics, it is better to add the details on the software and the tests used for the analysis.

The blood details such as blood group is also important in periodontitis. Hence in the Discussion, it is better to discuss regarding the blood groups.

https://www.researchgate.net/publication/339136933_Evaluation_of_Association_Between_the_Prevalence_and_Severity_of_Periodontal_Diseases_and_ABO_Blood_Groups_Among_Nepalese_Adults

https://www.ncbi.nlm.nih.gov/pmc/articles/PMC6420957/

6. Patents. Needs to be removed from the manuscript unless they have the details.

Minor English is needed with significant revision.

Author Response

We are grateful to the reviewers for their critical comments and useful suggestions that have helped to improve the manuscript. As indicated in the following responses, we have reflected all these changes in the revised version of our paper.

Response to Reviewer 1:

1) It is better to specify the age range rather than mentioning aged 20 years..

Response: We revised the methods as one more table (Table 1) was inserted to describe the characteristics of study participants including age distribution according to your suggestion.

Text Change: Data from adults aged 20 years of age or older from the National Health and Nutrition Examination Survey (NHANES) III in the United States were used.

2) Please mention the details of the study years

Response: We thought that the study years when we had conducted for several years could be omitted in most studies using secondary data.  Therefore, no additions were made. I hope you understand. Thanks.

Text Change:

3) The authors mention that they used the data from the National Health and Nutrition Examination Survey (NHANES) III in the United States. Did they obtain the approval?

Response: We found that in 2003, the NHANES Institutional Review Board (IRB) changed its name to the NCHS Research Ethics Review Board (ERB). In 2018, the name was changed from NCHS Research Ethics Review Board to NCHS Ethics Review Board. (https://www.cdc.gov/nchs/nhanes/irba98.htm)

Text Change:    Publicly available NHANES data sets have been previously approved by the National Center for Health Statistics Institutional Review Board, Center for Disease Control and Prevention; thus, no further internal ethical review was necessary.

4) In the statistics, it is better to add the details on the software and the tests used for the analysis.

Response: Thanks for the suggestion. We revised the methods as follows according to your suggestion.

Text Change: NHANES III used a complex, stratified, and multistage sampling design and was conducted by the National Center for Health Statistics (Hyattsville, MD, USA). SAS (version 9.1, SAS Institute Inc., Carey, NC, USA) survey procedures with weighting were used to consider national sampling, resulting in less bias in the strengths of association and standard errors observed.  ~~~ To evaluate effect modifications between periodontal disease, central adiposity, and components of metabolic syndrome, the logistic regression models including periodontal disease measures and metabolic syndrome components were initially stratified by central adiposity. Next, multiplicative terms between the categories of periodontal disease and central adiposity were introduced in the multivariate models to evaluate joint associations between categories

5) The blood details such as blood group is also important in periodontitis. Hence in the Discussion, it is better to discuss regarding the blood groups.

Response: We revised the discussion section as follows according to your suggestion.

Text Change: Fourth, an interesting finding of association between blood type and chronic periodontitis has been reported recently [42]. The present study did not consider the blood group as a confounder. This is one of limitations of this study.

  1. Mostafa D.; Elkhatat E. I.; Koppolu P.; Mahgoub M.; Dhaifullah E.; Hassan A. H. Correlation of ABO Blood Groups and Rh factor with the severity of generalized chronic periodontitis: Across sectional study in Riyadh, Saudi Arabia. Open Access Maced J Med Sci. 2019, 7, 617. Doi: 10.3889/oamjms.2019.044.

6) Patents. Needs to be removed from the manuscript unless they have the details.

Response: Deleted.

Reviewer 2 Report

I would like to thank you for the opportunity since I feel very fortunate to be able to review this article and I would like to congratulate the authors for this work. For me this topic is very important and has a lot of value. My suggestions are detailed below and my consideration at the end.

This manuscript investigated the association of periodontal measurements of clinical attachment loss and pocket depth with components of the metabolic syndrome. In addition, the epidemiologic role of central and total body adiposity in the relationship of periodontitis and hyperglycemia to other components of the metabolic syndrome was explored.

Title: The title is concrete, representative and indicative of the problem investigated in the manuscript. As a suggestion, the title should provide information about the place where the research was performed and provide information about the group of subjects.

Abstract: The abstract is clear and complies with the general rules for writing a good abstract. However, I would like to see a better description of the sample, indicating the context.  This is the most important section of the paper since it will be read many more times than even the manuscript itself, so it needs the most attention. A brief note on the importance of the research is an excellent ending to a high-level abstract.

Introduction

As I mentioned, I find this research extremely important in contributing to the field of health care. I do not disagree with the authors' justifications and read many very good and current arguments.  However, it is suggested to provide more information about metabolic syndromes.

It is suggested to the authors that based on the stated objective they highlight the research questions that help to conduct the research and discussion based on the findings found in which the study variables, the study population, and the expected result appear.

Material and method.

Participants. This section should be better defined. In this section (participants), the characteristics of the sample should be better described. It would be appreciated to include a table to help characterize the sample in the results section. 

Instruments and measures: Why was BMI used as an indicator and not ICC?

Statistical analysis. It is not reported whether the distribution of the data met the assumption of normality Were any tests performed for Why were regression models used and not Spearman's rho or Pearson's r (as appropriate) The choice of tests should be justified. 

Results: 

The results are displayed correctly and are easy to read and straightforward for a scholar unaccustomed to quantitative methodology. 

Discussion: It seems to me that a great job has been done in comparing the findings with other studies. Congratulations. I congratulate the authors for the inclusion of the section on practical and theoretical implications to evaluate the scope of the research.

Conclusions: They are clear and provide an answer to the stated objectives. 

I suggest considering this article for publication following minor revisions.

Author Response

We are grateful to the reviewers for their critical comments and useful suggestions that have helped to improve the manuscript. As indicated in the following responses, we have reflected all these changes in the revised version of our paper.

Response to Reviewer 2:

1) Title: The title is concrete, representative and indicative of the problem investigated in the manuscript. As a suggestion, the title should provide information about the place where the research was performed and provide information about the group of subjects.

Response: We revised the title as follows according to your suggestion.

Text Change: Association of adiposity with periodontitis and metabolic syndrome :  from the Third National Health and Nutrition Examination Survey of United States

2) Abstract: The abstract is clear and complies with the general rules for writing a good abstract. However, I would like to see a better description of the sample, indicating the context.  This is the most important section of the paper since it will be read many more times than even the manuscript itself, so it needs the most attention. A brief note on the importance of the research is an excellent ending to a high-level abstract.

Response: Thanks for the suggestion. We revised the abstract.

Text Change: -   This study included data from the National Health and Nutrition Examination Survey III of America on 12,254 adults aged 20 years of age or older with a blood sample, anthropometric measurements, and a periodontal examination were included. Clinical periodontitis measurements, including CAL and PD, were classified into quintiles or quartiles and compared.  ~

3) Introduction

As I mentioned, I find this research extremely important in contributing to the field of health care. I do not disagree with the authors' justifications and read many very good and current arguments.  However, it is suggested to provide more information about metabolic syndromes.

Response: Thanks for the suggestion. We revised the introduction as added two latest metabolic syndromes-related papers.

Text Change: -   Periodontal disease is also associated with metabolic syndrome components [33-36]. But the relationship with adiposity needs to be studied using epidemiological methods.

-            35. Jepsen S.; Suvan J.; Deschner J. The association of periodontal diseases with metabolic syndrome and obesity. Periodontology 2000, 83, 125-153. Doi: 10.1111/prd.12326.

-            36. Naqvi A.R.; Brambila M.F.; Martínez G.; Chapa G.; Nares S. Dysregulation of human miRNAs and increased prevalence of HHV miRNAs in obese periodontitis subjects. Journal of clinical periodontology 2019, 46, 51-61. Doi: 10.1111/jcpe.13040

4) It is suggested to the authors that based on the stated objective they highlight the research questions that help to conduct the research and discussion based on the findings found in which the study variables, the study population, and the expected result appear.

Response:  
Thank you so much for your comment. We have read thoroughly this manuscript and tried to improve based on study participants, risk factors, and results.

5) Material and method.

Participants. This section should be better defined. In this section (participants), the characteristics of the sample should be better described. It would be appreciated to include a table to help characterize the sample in the results section.

Response: An additional table (Table 1) was inserted to better describe and understand the characteristics of the study participants.

Text Change: Table 1

6) Instruments and measures: Why was BMI used as an indicator and not ICC?

Response: BMI is a measure of body fat based on height and weight that applies to adult men and women. It was used as an indicator because it is widely used.